# Recurrent *PTPRT/JAK2* mutations in lung adenocarcinoma among African Americans

Khadijah A. Mitchell [1,9], Noah Nichols [1,9], Wei Tang [1], Jennifer Walling [2], Holly Stevenson [2], Marbin Pineda[2], Roxana Stefanescu[3], Daniel C. Edelman[2], Andrew T. Girvin[3], Adriana Zingone[1], Sanju Sinha[1,4], Elise Bowman[1], Emily L. Rossi[1], Rony F. Arauz [1], Yuelin Jack Zhu[2], Justin Lack[5,6], Elizabeth Weingartner[7], Josh Waterfall [2], Sharon R. Pine [8], John Simmons[7], Paul Meltzer[2] & Bríd M. Ryan [1*]

Reducing or eliminating persistent disparities in lung cancer incidence and survival has been challenging because our current understanding of lung cancer biology is derived primarily from populations of European descent. Here we show results from a targeted sequencing panel using NCI-MD Case Control Study patient samples and reveal a significantly higher prevalence of *PTPRT* and *JAK2* mutations in lung adenocarcinomas among African Americans compared with European Americans. This increase in mutation frequency was validated with independent WES data from the NCI-MD Case Control Study and TCGA. We find that patients carrying these mutations have a concomitant increase in IL-6/STAT3 signaling and miR-21 expression. Together, these findings suggest the identification of these potentially actionable mutations could have clinical significance for targeted therapy and the enrollment of minority populations in clinical trials.

[1] Laboratory of Human Carcinogenesis, Center for Cancer Research, National Cancer Institute, Bethesda, MD 20892, USA. [2] Genetics Branch, Center for Cancer Research, National Cancer Institute, Bethesda, MD 20892, USA. [3] Palantir Technologies, 1025 Thomas Jefferson St, Washington, DC 20007, USA. [4] Cancer Data Science Laboratory, Center for Cancer Research, National Cancer Institute, Bethesda, MD 20892, USA. [5] NIAID Collaborative Bioinformatics Resource, National Institute of Allergy and Infectious Diseases, National Institutes of Health, Bethesda, MD 20892, USA. [6] Advanced Biomedical Computational Science, Frederick National Laboratory for Cancer Research sponsored by the National Cancer Institute, Frederick, MD 21702, USA. [7] Personal Genome Diagnostics, Baltimore, MD 21124, USA. [8] Rutgers Cancer Institute of New Jersey, Robert Wood Johnson Medical School, Rutgers, The State University of New Jersey, New Brunswick, NJ 08854, USA. [9] These authors contributed equally: Khadijah A. Mitchell, Noah Nichols. *email: Brid.Ryan@nih.gov

Lung cancer is the leading cause of cancer-related death in the United States (U.S.) and the second most common form of cancer diagnosed in both men and women[1]. Since public health records began tracking differences in lung cancer incidence and mortality by racial and ethnic groups in the U.S., disparities between European Americans (EAs) and African Americans (AAs) have been identified[2,3]. Specifically, lung cancer incidence is higher in AAs, especially among men[1]. AAs also have the highest mortality rate and the lowest 5-year survival rate compared with other racial and ethnic groups[1]. The factors contributing to this health disparity are multifactorial[4]. For example, access to high quality health care is an important factor in lung cancer outcomes. In terms of incidence, it is likely that tobacco plays a role in the observed differences given that it is the leading etiological exposure associated with the lung cancer development[4]. However, AAs have a lower tobacco consumption overall compared with EAs[5] and data show that the difference in lung cancer incidence persists at equal categories of cigarettes smoked per day[6]. This suggests a divergence in the etiology of lung cancer in the U.S. between racial and ethnic groups. As exposures are tightly linked with tumor biology[7], it is possible that such differences in disease etiology could be reflected at the genomic level.

Our current understanding of lung cancer biology is primarily derived from populations of European descent. Given the persistent disparities that exist in lung cancer incidence and survival between AAs and EAs, it is important to characterize tumor biology across racial and ethnic groups. Large-scale genomic studies have highlighted genetic heterogeneity in lung cancer[8–10]. By identifying driver mutations, these studies have greatly contributed to the development of targeted pharmacological drugs for the treatment of cancer, and, through the ability to detect circulating tumor DNA, are also being leveraged for early diagnostics[11]. To date, few studies have investigated the somatic mutation landscape of lung cancer in AAs, and of those that have, the studies often included a small panel of genes or focused on hotspot mutations; others have focused on tumor tissue only[12–15]. Here, we report two genes, *PTPRT* and *JAK2*, that are recurrently mutated in lung adenocarcinoma (LUAD) among AAs.

## Results

**AAs have a complex lung cancer mutational landscape.** We conducted targeted exome sequencing of 129 tumor/adjacent non-involved pairs of fresh-frozen tissue from self-reported AAs (Supplementary Table 1) in the NCI-MD Case Control Study. Admixture analysis was consistent with self-reported race for 98% of the samples, comparable with The Cancer Genome Atlas (TCGA[16]; Supplementary Data 1). Of the 564 genes examined (Supplementary Data 2), 67 were not mutated in any of the patients (Supplementary Data 3) and 13 patients did not have mutations in the genes sequenced. We identified 4,136 somatic single-nucleotide variants (SNVs) and indel events (Supplementary Data 4; median/patient = 24, range = 0–426; Fig. 1a; Supplementary Data 4), reflecting the genetic heterogeneity of the population. As expected, tumors from smokers had more mutations than never smokers (average = 38, 37, and 5 for current, former and never smokers, respectively). The median number of mutations that passed the second filter, i.e., likely to alter protein function, was 14 (range = 0–132; Supplementary Data 5). Roughly a quarter (24%) of tumors did not harbor a mutation in the Oncovar gene panel, which is consistent with the previous observations[8,10,17]. It is possible that other somatic copy number-based genomic events, rare driver mutations, or epigenomic changes drive carcinogenesis in these tumors. Using a recent definition for hypermutation (>10 somatic SNVs/megabase (Mb))[18], 59 samples were classified as hypermutated. The patient

with the highest mutation burden was a current smoker with 64 pack-years of tobacco smoke consumption, who presented with adenocarcinoma. Known DNA repair genes—*XRCC1*, *FANCA*, *BRCA1*, *PARP1*, and *ERCC4*—were mutated and a somatic mutation signature consistent with defects in mismatch repair (signature 20) were observed in hypermutated patients (Supplementary Fig. 1). Mutations in mismatch repair genes have been associated with a hypermutated phenotype. *MSH2*, *MSH6*, *MLH1*, and *PMS2* were included in the gene panel and only one of the patients with a hypermutated tumor had a mutation, which was a missense R638S mutation in *MSH2*.

As expected[19], the most common nucleotide change was a C > A transversion (Supplementary Fig. 2a). Each tumor somatic profile was further contextualized in terms of known mutational signatures[20]. Eleven dominant mutational signatures were observed across AA lung cancers (Fig. 1b; Supplementary Data 6). Consistent with previous work[21], signature 4 was the main signature observed in both LUAD and lung squamous cell carcinoma (LUSC) tumors from AAs and associated with smoking exposure (Fig. 1b). The APOBEC signatures 2 and 13 were also observed. Mutational signatures 3 (homologous recombination deficiency), 18 (potentially due to reactive oxygen-species-induced DNA damage)[22], and 24 (which, like 4, has a C > A bias, is associated with aflatoxin exposure; Fig. 1b; Supplementary Data 6) were also observed in many of the AA tumor samples. However, as this was a targeted gene panel, future studies should conduct a more thorough study with whole-exome sequencing (WES).

Consistent with published studies, we observed a complex mutational landscape of lung cancer in AAs with considerable heterogeneity in the somatic landscape between individuals and little evidence for dominant driver genes (Supplementary Fig. 1c). We confirmed genomic alterations previously identified in lung cancer including *TP53*, which was the most mutated gene (Fig. 1c).

**AAs have high *PTPRT* and *JAK2* mutation frequencies.** As the frequency of somatic mutations varies by histological subtype, we report mutation frequencies for LUAD and LUSC separately. Fifteen genes were significantly mutated in LUSC (Supplementary Fig. 2b; FDR $P < 0.1$). For most of these genes, the mutation frequency was comparable among AAs and EAs (Supplementary Data 7). In LUAD, 18/54 samples (33%) did not have a significantly recurrent mutation compared with 14/52 (27%) in LUSC, suggesting that, as in EAs, a large proportion of genomic drivers of lung cancer remain to be identified.

Eleven genes were significantly mutated in LUAD (Supplementary Fig. 2c, d). A comparison of driver genes between AAs and EAs (TCGA) shows that the global pattern of driver gene mutation frequencies is generally similar between EAs and AAs (Supplementary Data 7). However, *STK11* and *RB1* mutations occurred in 19% and 11% of LUAD tumors among AAs, respectively, which is higher than the frequency reported for EA patients in TCGA[9] (Supplementary Fig. 2c). *TP53* mutations were slightly higher among AAs compared with EAs, consistent with previous observations[16].

We further found that the frequency of mutations in *PTPRT* and *JAK2* are higher in AAs compared with EAs (Fig. 2a). Our data indicate that 13/54 (24%) of LUAD patients have mutations in *PTPRT* and that 4/54 (7.4%) have mutations in *JAK2*, compared with 8% and 2% in EAs, respectively (Fig. 2a). These mutations do not tend to co-occur in the same patient. Specifically, of the 15 patient samples (all histology combined) that carried a mutation in *PTPRT* and the 11 that carried a mutation in *JAK2*, only 1 sample had a mutation in both genes

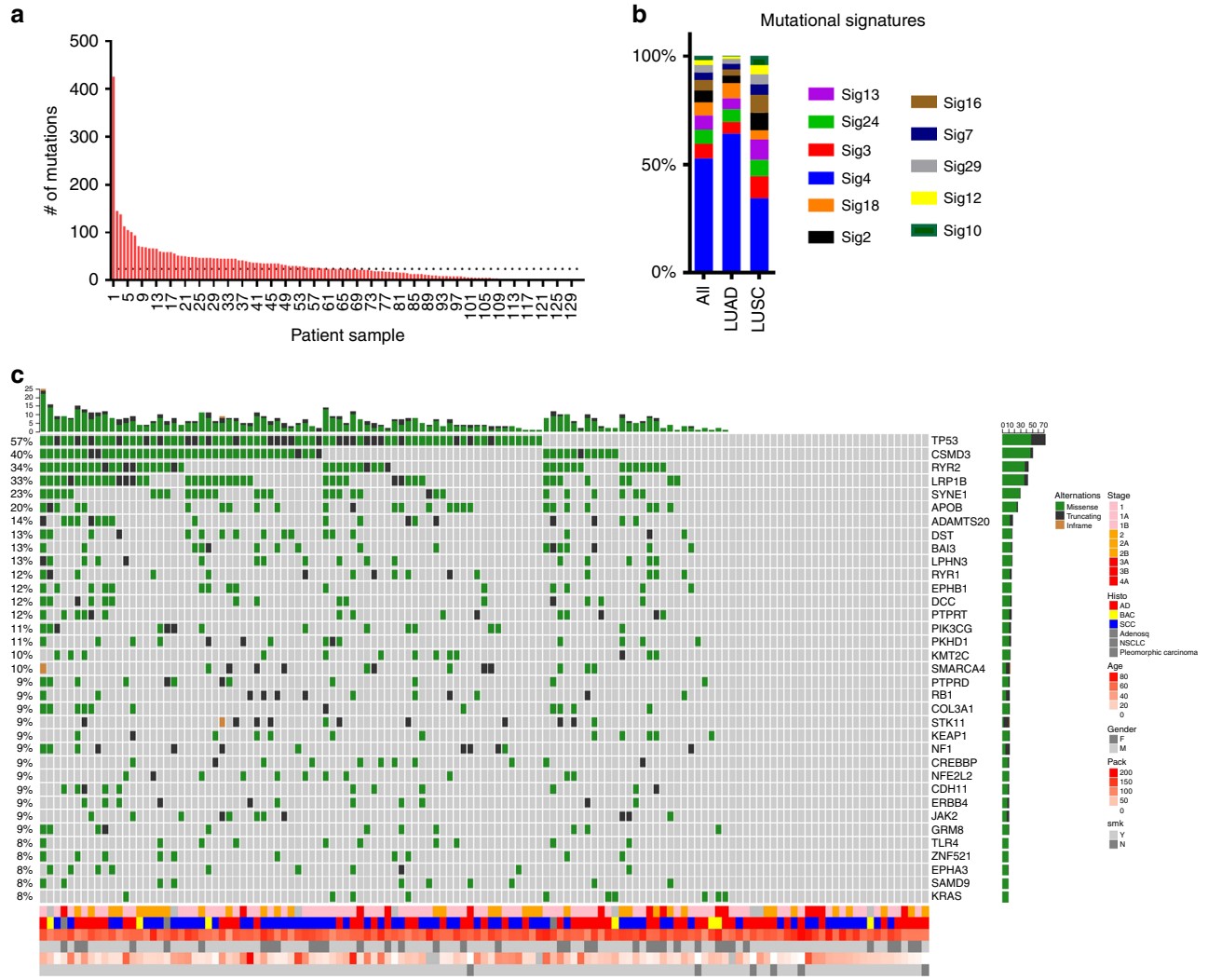

**Fig. 1 Somatic mutation profile of lung cancer in AAs. a** Distribution of somatic mutation number across patient samples. **b** Summary of mutational signatures in LUAD and LUSC samples from the NCI-MD Case Control Study. **c** Oncoprint outlining the co-occurrence of somatic mutations in all samples in the NCI-MD Case Control Study (histo, histology; AD, adenocarcinoma; BAC, bronchioalveolar carcinoma; SCC, squamous cell carcinoma; adenosq, adenosquamous cell carcinoma; pack, pack-years of smoking; smk, smoking status). Dashed line indicates the median.

(two-sided Fisher's exact test $P < 0.001$). In LUAD, no sample carried a mutation in both genes suggesting that these mutations are mutually exclusive (two-sided Fisher's exact test $P = 0.001$). *PTPRT* was not mutually exclusive of other known key oncogenes and tumor suppressors (Supplementary Data 8).

Combined, *PTPRT* and *JAK2* are mutated in >30% of tumors from AAs and ~10% of tumors from EAs (Fig. 2b). To validate these observations, we first used data from TCGA (Supplementary Data 1) and replicated the statistically higher frequency of *PTPRT* (AA 20%, EA 8%, two sample test of proportions $P = 0.0004$) and *JAK2* (AA 6%, EA 2%, $P = 0.025$) mutations in LUAD from AAs (Fig. 2a; Supplementary Data 7). Secondly, we conducted WES on an additional independent set of 50 tumor and normal pairs from AAs and EAs in the NCI-MD Case Control Study (Supplementary Data 1, 2 and 9). Again, we observed a higher frequency of *PTPRT* (AAs 21%, EAs 9.6%, two sample test of proportions $P = 0.014$) and *JAK2* (AAs 10%, EAs 0%, two sample test of proportions $P = 0.08$) mutations in tumors from AAs (Fig. 2a). Similar to data in lung cancer among EAs and other cancer types[23], there were no clear hotspot mutations and the mutated codons were spread throughout *PTPRT*, including the phosphatase and extracellular domains (Fig. 2c). To our

knowledge, this is the first time this observation has been reported in AAs. Previous studies based on targeted sequencing panels did not include *PTPRT*[12–14], which likely explains why this observation was not reported before. Also, our inclusion of matched normal samples indicates that the events are somatic and not germline, which is an important observation given to the recent description that 10% of the pan-African genome is not represented in the current reference genome[24].

PTPRT and JAK2 function downstream of cytokine and interferon signaling to regulate STAT3[23], which is an oncogenic driver and hallmark of cancer[25]. Integrating total RNAseq data for 23 samples for which we had both targeted exome sequencing and RNAseq data ($n = 6$ mutant and 17 wild type), we observed an enrichment of IL6/JAK2/STAT3 and interferon signaling among lung tumors carrying either *PTPRT* or *JAK2* mutations (Fig. 2d). We also observed an enrichment of PI3K signaling, consistent with the literature[26]. We then analyzed microRNA (miRNA) transcriptional targets of STAT3[27], and observed increased miR-21 (Fig. 2e) and miR-181b (Supplementary Fig. 2) in tumor samples carrying mutations in *PTPRT* or *JAK2*, while non STAT3 targets, such as miR-126, were similar (Supplementary Fig. 3). These data suggest that an increased frequency of loss

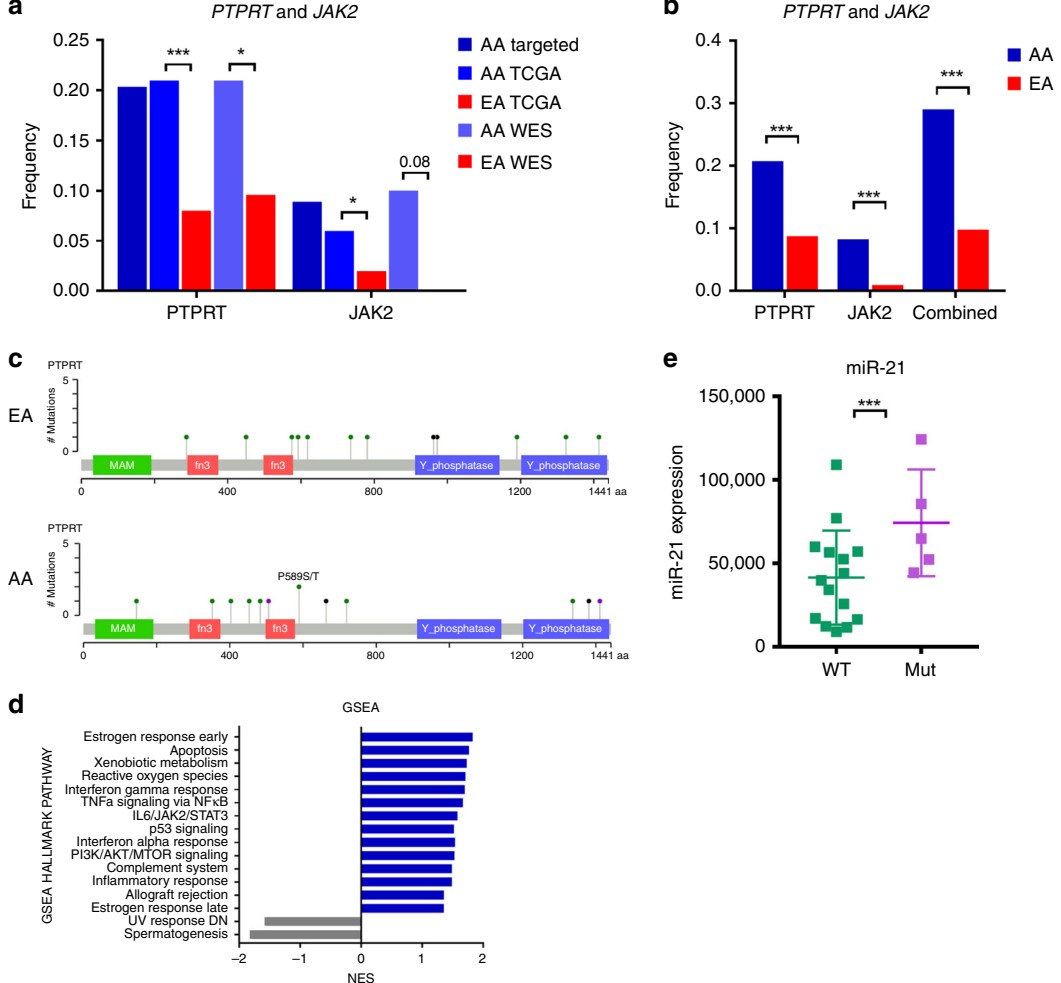

**Fig. 2 *PTPRT* and *JAK2* mutations in LUAD. a** *PTPRT* and *JAK2* mutations in the NCI-MD Case Control Study using targeted sequencing and WES, and in TCGA using WES and **b** combined. **c** Graphical distribution of individual mutations in *PTPRT* in EAs and AAs. **d** GSEA of gene expression changes in *PTPRT* and *JAK2* mutant samples compared with wild type. **e** Levels of miR-21 in *PTPRT* and *JAK2* mutant samples compared with wild type. Error bars indicate the s.d. *$P < 0.05$, **$P < 0.01$, ***$P < 0.001$, two-sided Student's $t$-test. Source data are provided as a Source Data file. WES, whole-exome sequencing; AA, African American, EA, European American; TCGA, The Cancer Genome Atlas; GSEA, Gene set enrichment analysis.

of function *PTPRT* and *JAK2* mutations may drive STAT3 activity in subsets of non-small cell lung cancer (NSCLC) that are enriched among AAs.

## Discussion

We report the somatic mutation profiles of 129 matched lung cancers from AAs across the coding regions of 564 pan-cancer genes (and six whole gene regions) and confirm key findings with data from (1) TCGA and (2) WES of 50 EAs and AAs. Roughly, a quarter (24%) of the tumors in our analysis did not harbor a mutation in the Oncovar gene panel, which is consistent with the previous observations[8,10,17]. It is possible that other somatic copy number-based genomic events, rare driver mutations, or epigenomic changes drive carcinogenesis in these tumors. We did not observe substantial differences in the mutation frequency of known driver genes according to ancestry in either LUAD or LUSC. However, we identified an increased prevalence of *PTPRT* and *JAK2* mutations in LUAD from AAs. We validated this observation using whole-exome data from both TCGA and an independent set of samples from NCI-MD. Combined, ~30% of tumors from AAs carried mutations in *PTPRT* and/or *JAK2* genes compared with 10% of EAs. To our knowledge, this is the first

time this observation has been reported in AAs. Other protein phosphatases mutated in cancer, e.g., *PTPRD*, also negatively regulate STAT3 activation. A comprehensive study on the mutation frequency of these phosphatases and other STAT3 pathway regulators in LUAD from AAs is also warranted[28].

TCGA has reported a fusion partner of *PTPRT* in lung cancer, *EXD2*. Therefore, although calling fusion genes from WES data can be problematic and error prone[29], if searching for a specific gene the likelihood of false positive findings can be reduced. We detected putative *PTPRT* fusion genes in nine samples, though none had a similar partner gene or the same partner as previous reported fusions in TCGA. Further, as the minor allele fraction for either split reads or spanning pairs is very low in our study (Supplementary Data 10), it suggests that these subclonal fusions are not pathogenic or biologically relevant. Because fusion events that result in a well-expressed transcript are more easily and more reliably detectable from RNAseq data, future studies with RNA-seq data should explore whether these putative fusion genes manifest as transcribed variants.

Interestingly, our recent work demonstrated that while IL-6 is associated with lung cancer diagnosis in both EAs[30] and AAs[31], the effect size was considerably larger among AAs, which is further evidence that this IL-6/JAK2/STAT3 pathway is

important among AAs. We hypothesize that patients with *PTPRT* and *JAK2* mutations could be candidates for targeted therapy and as such, our findings have implications for the recruitment of patients into clinical trials. For example, the initial conception to use JAKs as therapeutic targets was based on the identification of an activating mutation in *JAK2* linked to myeloproliferative neoplasms[32]. The rationale for their use in these disorders has also been linked with perturbed JAK/STAT signaling, either due to somatic mutations or transcriptomic changes[33]. Recent work by Pitroda and colleagues found that a selective JAK2 inhibitor is cytotoxic to NSCLC cells in the context of constitutive IFN-stimulated JAK/STAT gene expression and that tumor cell-intrinsic expression of IFN-inducible PD-L1 was abrogated by the selective inhibitor[34]. In fact, somatic *JAK1/2* mutations were shown to mediate primary resistance to PD-1 blockade because of an inability to signal through the interferon gamma receptor pathway, making it possible that patients harboring such mutations would be unlikely to respond to PD-1 blockade therapy[35]. Taken together, these findings suggest a potential role for JAK2 inhibitors in lung cancer in the context of a specific genomic background that could also possibly work in tandem with immune checkpoint inhibition.

Current JAK inhibitors are not always selective and most do not target specific mutations, though newer generations of JAK inhibitors demonstrate selective inhibition. JAK2 inhibitors might not work in *PTPRT* mutant tumors because other JAKs can, in theory, activate STAT3. As such, STAT3 inhibitors are good candidates for the tumors, we describe in our study. Interestingly, we conducted an agnostic analysis of differential drug sensitivity among cell lines mutant for *JAK2* or *PTPRT* using the depmap database[36] [https://depmap.org] and identified a STAT inhibitor with selective growth inhibition in *PTPRT* mutant cells (Supplementary Data 11). Our findings therefore raise the hypothesis that patients carrying these mutations may be more likely to respond to drugs that target this pathway than patients without these mutations. However, detailed mechanistic experiments will be needed to determine whether these are indeed actionable mutations, especially given a recent report that up to half of *JAK2* mutations in nonsmall cell lung cancer can be inactivating[37].

Our study has several strengths. It uses fresh-frozen tissues and matched tumor and non involved adjacent tissues. This study design gives us the ability to call true somatic mutations and is especially important in light of recent findings showing that up to 10% of the genome in individuals of African ancestry are not captured, using the current reference genome[24]. Most of these differences map to intergenic and noncoding regions, as such, their impact on a targeted exome-sequencing panel would be expected to be limited in nature. However, future work should address these novel genomic sequences and assess them for potential health-associated variants. Second, we used two additional datasets to confirm our results. As TCGA includes participants from across the U.S. and our samples were from the Baltimore region of Maryland, leveraging the TCGA database allowed us to compare our results to AAs from across the U.S. Whether or not population differences in *PTPRT/JAK2* mutations extend to populations of Asian descent, or indeed other minority and under-represented populations, remains to be determined. TCGA has eight LUAD patients classified as Asian, one (12.5%) of which carries a *PTPRT* mutation, suggesting that the frequency in Asian populations is more closely aligned with EAs.

In summary, we show that the global frequency of somatic mutations is similar in tumors from EAs and AAs. However, we present evidence that somatic mutations in *PTPRT* and *JAK2* are enriched in AAs and hypothesize that these mutations may be actionable. As this is a putatively targetable pathway, preclinical studies are needed to determine whether tumors carrying these

mutations affect outcome or response to therapy directed against IL-6/JAK2/STAT3 signaling.

## Methods

**Patient samples and DNA extraction.** Patients were selected from an ongoing case control study conducted by the NCI and the University of Maryland (Supplementary Data 1). This NCI-MD Case Control Study was conducted in accordance with the Declaration of Helsinki. Institutional review board approval was granted from NCI and participating hospitals and registered on clinicaltrials.gov [https://clinicaltrials.gov/ct2/show/NCT00339859]. Written informed consent was obtained from all patients. Patients for this study were recruited between 1984 and 2013. At the time of surgery, a portion of the tumor specimen and non involved adjacent lung tissue was flash frozen and stored at −80 °C until needed. Clinical and pathological information was obtained from medical records, tumor boards, and pathology reports.

Total genomic DNA was extracted using DNeasy Blood and Tissue Kit (QIAGEN, Valencia, CA). DNA quality and yield were determined using a NanoDrop Spectrophotometer (Thermo Fisher Scientific, Wilmington, DE). The initial study population included 141 tumor–normal pairs. One sample failed QC and was not suitable for sequencing. Four samples had poor quality normal tissue and were excluded due to the inability to match with tumor tissue. After sequencing was complete, seven samples were excluded due to quality of sequencing data. Thus, in total, 12 samples were excluded and the final study cohort consisted of 129 tumor–normal pairs.

The validation study population included an independent sample set ($n = 50$ samples) from the same ongoing case-control study. DNA was extracted from 15 μm sections of FFPE tissue using the Qiagen DNA FFPE Tissue Kit. Input for library prep was 500 ng.

**Targeted exome sequencing and data processing.** Simultaneous fragmentation and adaptor ligation was performed on input gDNA (50 ng) by tagmentation, using the Nextera DNA Library Preparation kit, according to the manufacturer's protocol (Illumina). Products with a mean size of 350 bp +/−20% were purified using the Agencourt AmpureXP Purification System (Beckman Coulter). Amplification and dual indexing of purified samples was performed using Illumina PCR primers InPE1.0 and InPE2.0, and primer indices (8 bp). Hybridization capture of pooled indexed libraries was performed according to the manufacturer's protocol using NCI Oncovar V4, an Agilent SureSelect Custom DNA kit (Agilent Technologies) targeting 2.93 Mb of exonic sequence in 564 genes found to be mutated in diverse solid tumors (Supplementary Data 2) with full coverage of six genes (*CDKN2A*, *PTEN*, *SDHA*, *SDHC*, *TP53*, and *VHL*)[38]. In addition, xGen Blocking Oligos (Integrated DNA Technologies Inc., Coralville, IA) specific to Nextera library adaptor sequences were used during hybridization according to manufacturer's recommendations. The libraries were sequenced on an Illumina NextSeq 500 or HiSeq 2500 instrument by paired-end $2 \times 75$ bp to an average target region depth of ~140×. Alignments to the hg19 human reference genome assembly were performed with BWA-MEM (release 0.7.10, July 13 2014, r789)[39], indel realignment by GATK IndelRealigner (version 3.4-0- g7e26428)[40], and duplicates were marked with picard MarkDuplicates (version 1.129)[41]. Somatic SNVs and small insertions and deletions were called with Strelka 2.0.17[42]. All variants are reported as filter 1, while those mutations likely to alter protein function, i.e., nonsynonymous, frame shift, splice site, start/stop site SNVs, and codon insertion or deletions, are reported as filter 2. Sequencing statistics are reported in Supplementary Data 12. A two-sample test of proportions was used to analyze statistical differences in the frequency of somatic mutations between populations.

**WES and data processing.** WES was performed at Personal Genome Diagnostics (Baltimore, MD)[43]. In brief, DNA was extracted from FFPE tissue and matched blood or saliva samples, using the Qiagen DNA FFPE Tissue Kit or Qiagen DNA Blood Mini Kit (Qiagen). Genomic DNA from tumor and normal samples was fragmented and used for Illumina TruSeq library construction (Illumina), according to the manufacturer's instructions. Briefly, 500 ng of genomic DNA in 100 ml of TE (tris-EDTA) was fragmented and purified using Agencourt AMPure XP beads (Beckman Coulter). Exonic regions were captured in solution using the Agilent SureSelect kit (Agilent). PE sequencing, resulting in 100 bases from each end of the fragments for exome libraries, was performed using Illumina instrumentation (Illumina). Sequence reads were aligned against the human reference genome (version hg19). Somatic mutations were identified using VariantDx and Cerebro custom software[43,44]. Fusion genes were called with the Manta program[45].

**Mutation calls in TCGA.** Somatic mutations calls for LUAD and LUSC were downloaded from Firehose for EAs and AAs separately (Supplementary Data 7).

**Mutation significance analysis.** Mutation significance was performed using the MutSig2CV algorithm [https://software.broadinstitute.org/cancer/cga/mutsig_run]. The current version improves the background mutation rate estimation by pooling data from neighbor genes in covariate space, and substantially reduces the number of false–positive findings. Tables with mutation data, per-sample coverage, gene

covariables, and mutation type were imported to the software. Genes with a Bonferroni-corrected $P < 0.05$ are considered significant[46].

**Mutational signature analysis**. Mutational signatures in the targeted sequencing data were analyzed using R/Bioconductor package "MutationalPatterns". The package covers a wide range of tools including: mutational signatures, transcriptional and replicative strand bias, genomic distribution, and association with genomic features. References mutation signature were obtained from the COSMIC website [https://cancer.sanger.ac.uk/cosmic/signatures] for 30 signatures. The current signatures were then determined by the contribution of 30 known mutational signatures on a single sample level by finding the optimal linear combination of mutational signatures that most closely reconstructs the mutation matrix[47].

**Measurement of miR-21**. miRNA expression for miR-21 in lung adenocarcinoma tumor and normal tissues was extracted from our previous Nanostring analysis of miR-21, and compared between *PTPRT*- and *JAK2*-mutated samples ($n = 4$) and wild-type samples ($n = 9$). The miRNA microarray data discussed in this publication have been deposited in National Center for Biotechnology Information's GEO and are accessible through GEO Series accession number GSE63805. Tests for statistical differences in miR-21 expression between mutated and nonmutated samples were conducted using two-sided Student's *t*-test.

**Gene set enrichment analysis**. We integrated total RNAseq data for 23 samples, where we had both mutation and RNAseq data ($n = 6$ mutant and 17 wild type) in the NCI-MD study using the Palantir Foundry platform. Genes with fewer than 1 read per million in at least three members of each group were removed. Following quantile normalization and differential expression analysis using the R/Bioconductor package limma, gene set enrichment analysis (GSEA) was performed using the fgsea package and the MSigDB Hallmark Pathways.

**Genetic ancestry**. For admixture analysis, we utilized the 1000 Genomes Project phase III[48] superpopulations as reference populations, where we removed rare variants (i.e., <5% across all of the phase III 1000 genomes), all INDELs and any SNPs that were not biallelic. We then used the tool Admixture v1.3.0[49] to estimate ancestry proportions for each of the 1000 Genomes Project superpopulations.

## Data availability

The datasets generated during the current study have been uploaded to the dbGaP repository in compliance with the NIH Genomic Data Sharing Policy. Data can be accessed at [https://www.ncbi.nlm.nih.gov/projects/gap/cgi-bin/study.cgi?study_id=phs001895.v1.p1]. Raw data for Figs. 1 and 2, and Supplementary Figs. 1–3 are provided in the Source Data File.

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

## Acknowledgements

This study was supported by the NIH Intramural Research Program at the National Cancer Institute.

## Author contributions

These authors contributed equally: Khadijah A. Mitchell and Noah Nichols. Conception and design: K.A.M, N.N., and B.M.R. Development of methodology: D.C.E., J.W., P.M., J.S., and E.W. Acquisition of data (provided animals, acquired and managed patients, provided facilities, etc.): K.A.M, A.Z., B.M.R., E.B., D.C.E., P.M., J.W., H.S., M.P., J.S., and E.W. Analysis and interpretation of data (e.g., statistical analysis, biostatistics, and computational analysis): K.A.M., N.N., W.T., R.S., A.T.G., S.S., D.C.E., Y.J.Z., J.L., J.W., S.P., J.S., P.M., and B.M.R. Writing, review, and/or revision of the manuscript: K.A.M., N.N., W.T., R.S., A.T.G., S.S., Y.J.Z., J.L., J.W., S.P., E.W., J.S., P.M., B.M.R., J.W., H.S., M.P., D.C.E., A.Z., E.B., E.R., and R.F.A. Administrative, technical, or material support (i.e., reporting or organizing data, constructing databases): K.A.M, N.N., and B.M.R. Study supervision: B.M.R.

## Competing interests

The authors declare no competing interests.
