## [Peer Review File · Nature Communications]

Reviewers' comments:

Reviewer #1 (Remarks to the Author):

This is a very interesting study indicating that lung adenocarcinoma of African Americans has higher mutation frequencies of PTPRT and JAK2 mutations than European Americans. The authors analyzed frozen tissue samples, therefore, the results are reliable. The reviewer has several concerns for the manuscript.

1. Gene fusion, a major driver alteration, has not been analyzed, so, it is ambiguous whether PTPRT and JAK2 are playing a driver role in lung carcinogenesis or not. Mutually exclusiveness between PTPRT/JAK2 mutations and known driver oncogene alterations should be shown.
2. Experiments using lung cancer cell lines harboring JAK2/PTPRT mutations should be done to show that JAK2/PTPRT mutations are actionable.
3. Information on other populations, such as Asians, has not been included, so, the comprehensive picture of difference in gene alterations according to population is unclear; i.e., the authors simply showed a difference between African Americans and European Americans.
4. The reviewer does not fully agree that African Americans are called as a minority.

.

Reviewer #2 (Remarks to the Author):

This study reports some distinct alterations in PTPRT/JAK2 mutation in lung adenocarcinoma among African Americans. The strengths are a) study focused on a population often overlooked in genomic studies b) confirmation from two different cohorts including one from TCGA. I have a few comments

1. The abstract should include more details on the initial cohort and validation cohort

2. The text should highlight whether mismatch repair genes were altered in the hypermutated samples
3. It is important to be careful in discussing inter tumor heterogeneity when multiple samples from a given lesion have not been studied and the panel of genes tested was limited at least in the initial discovery cohort (page 4)
4. I would delete the part discussing CSMD3, RYR2 and SYNE1- they are not considered relevant in this context as the authors state
5. in the absence of functional studies, it is a leap of faith to say that "our findings suggest that patients carrying these mutations may be more likely to respond to such drugs"- This sentence should be modified
6. It is not correct to state this is the first study to examine genomic alterations in AA
7. There is not much discussion on the alterations (if any) that are unique to African ancestry not captured in the current reference genome

Response to Reviewers

Reviewer #1 (Remarks to the Author):

This is a very interesting study indicating that lung adenocarcinoma of African Americans has higher mutation frequencies of PTPRT and JAK2 mutations than European Americans. The authors analyzed frozen tissue samples, therefore, the results are reliable. The reviewer has several concerns for the manuscript.

1. Gene fusion, a major driver alteration, has not been analyzed, so, it is ambiguous whether PTPRT and JAK2 are playing a driver role in lung carcinogenesis or not. Mutually exclusiveness between PTPRT/JAK2 mutations and known driver oncogene alterations should be shown.

We thank the reviewer for raising the important point of mutual exclusivity, especially when it relates to putative oncogenes. In our study, *JAK2* and *PTPRT* mutations were primarily mutually exclusive of each other. Of the 15 NSCLC patient samples that carried a *PTPRT* mutation and 11 that carried a *JAK2* mutation, only 1 sample had a mutation in both genes. Among the LUAD samples, where we saw enrichment of *PTPRT* and *JAK2* mutations among African Americans, the mutations were mutually exclusive. On page 7, we have modified the text to make this clearer. We have also added a supplementary table (9) to show these data, along with data showing the mutual exclusive relationship between *JAK2/PTPRT* and other driver genes.

The reviewer also mentioned that we did not analyse fusion genes. We are not able to do this on the targeted gene panel, but it is, in theory, possible with the WES data. However, structural variant calling is not reliable with short read sequencing, especially in WES data¹. Fusion events that result in a well-expressed transcript are more easily and more reliably detectable from RNA-seq data. However, if one has a key gene in mind, such as *PTPRT*, it is somewhat possible to confirm it with WES data. As one of the EA samples in TCGA had a *PTPRT-EXD2* fusion that was predicted to be oncogenic, we looked for this in our WES data using the Manta pipeline. We found that some samples had putative *PRPRT* fusion genes, but none was with a consistent partner or with *EXD2*, as seen in TCGA. Further, fusion genes of *PTPRT* (n=4) were also detected in breast invasive carcinoma in TCGA data. Of these four, one was in a tumor from an African American patient. All these fusion partners were different and none overlapped with the fusion partners found in our study. We agree with the reviewer that these are important points to add, and therefore we have added these data now to the manuscript.

Page 6:

Specifically, of the 15 patient samples (all histology combined) that carried a mutation in PTPRT and the 11 that carried a mutation in JAK2, only one sample had a mutation in both genes (Fisher's exact test $P < 0.001$). In LUAD, no sample carried a mutation in both genes suggesting that these mutations are mutually exclusive (Fisher's exact test $P = 0.001$). PTPRT was not mutually exclusive of other known key oncogenes and tumor suppressors (Supplementary Table 9).

Page 9:

TCGA has reported a fusion partner of PTPRT in lung cancer, EXD2. Therefore, although calling fusion genes from WES data can be problematic and error prone¹, if searching for a specific gene the likelihood of false positive findings can be reduced. We detected putative PTPRT fusion genes in nine samples, though none had a similar partner gene or the same partner as previous reported fusions in TCGA. Further, as the minor allele fraction for either split reads or spanning pairs was very low (Supplementary Table 9), it suggests that these sub-clonal fusions are not pathogenic or biologically relevant. Because fusion events that result in a well-expressed transcript are more easily and more reliably detectable from RNAseq data, future studies with RNAseq data should explore whether these putative fusion genes manifest as transcribed variants.

2. Experiments using lung cancer cell lines harboring JAK2/PTPRT mutations should be done to show that JAK2/PTPRT mutations are actionable.

We agree with the reviewer that in order to say that these are potentially actionable mutations experiments would be needed to verify it. At present, we have an R01 pending that would specifically address those questions and anticipate that it would take considerable time to complete them, i.e., more than a year. We have modified our language to make it clear that we hypothesize that these variants are actionable, rather than saying the data suggest that they are. We appreciate that it is important to articulate this important difference. Our integrative transcriptome work indicated that tumors carrying *PTPRT* or *JAK2* mutations have different tumor biology, at least as it pertains to the IL-6/JAK/STAT pathway.

Indeed, the initial conception to use JAKs as therapeutic targets was based on the identification of an activating mutation in *JAK2* linked to myeloproliferative neoplasms². The rationale for their use in these disorders has also been linked with perturbed JAK/STAT signaling, either due to somatic mutations or transcriptomic changes. Most anti-JAK drugs act as competitive inhibitors to JAK proteins. Further, the efficacy of IL-6 and IL-6R inhibitors for several immune-related diseases is considered to be linked with an underlying prominence of IL-6 pathway dysregulation. However, these inhibitors, and the pathways they target, can be pleiotropic. Targeting IL-6 for example, needs to consider classic signaling, trans-signaling and trans-presentation involving IL-6 itself, the receptor IL-6R, the binding receptor gp130 and soluble IL-6R. Current JAK inhibitors are not always selective, most do not target specific mutations, though newer generations of JAK inhibitors are demonstrating select inhibition. Further, in general, JAK2 inhibitors have failed in solid tumors, partly because JAK2 inhibitors have off-target effects and are not selective against JAK2. It is therefore likely that more efforts are needed to develop better strategies to target this pathway in solid tumors. Moreover, JAK2 inhibition might not work in *PTPRT* mutant tumors because other JAKs can in theory activate STAT3. As such, STAT3 inhibitors are potentially good candidates for the *PTPRT*-mutant tumors we describe in our study. As an intracellular molecule, it is difficult to target, though the recent identification of “stattic”, a SH2 domain targeting compound identified via a small molecule screen, bears some promise³.

To generate data supporting our hypothesis, we leveraged the recently collated ‘depmap’⁴ study to agnostically test whether cell lines harboring mutations in *JAK2* or *PTPRT* would have

differential drug sensitivity. Interestingly, a JAK2 inhibitor and a STAT inhibitor were identified in the analysis, respectively, and support the hypothesis that these mutations could be actionable. We have now included these data as supplemental table 11. On page 10, we have modified the text to reflect these important points raised by the reviewer and added additional supporting data.

Page 9/10

Interestingly, our recent work demonstrated that while IL-6 is associated with lung cancer diagnosis in both EAs⁵ and AAs⁶, the effect size was considerably larger among AAs, which is further evidence that this IL-6/JAK2/STAT3 pathway is important among AAs. We hypothesize that patients with PTPRT and JAK2 mutations could be candidates for targeted therapy and as such, our findings have implications for the recruitment of patients into clinical trials. For example, the initial conception to use JAKs as therapeutic targets was based on the identification of an activating mutation in JAK2 linked to myeloproliferative neoplasms². The rationale for their use in these disorders has also been linked with perturbed JAK/STAT signaling, either due to somatic mutations or transcriptomic changes⁷. Current JAK inhibitors are not always selective and most do not target specific mutations, though newer generations of JAK inhibitors demonstrate selective inhibition. JAK2 inhibitors might not be effective in PTPRT mutant tumors because other JAK proteins can, in theory, activate STAT3. As such, STAT3 inhibitors are good candidates for the PTPRT-mutant tumors we describe in our study. Interestingly, we conducted an agnostic analysis of differential drug sensitivity among cell lines mutant for JAK2 or PTPRT using the depmap database⁴ (<https://depmap.org>) and identified a STAT inhibitor with selective growth inhibition in PTPRT mutant cells (Supplementary Table 11). Our findings therefore raise the hypothesis that patients carrying these mutations may be more likely to respond to drugs that target this pathway than patients without these mutations. However, detailed mechanistic experiments will be needed to determine whether these are indeed actionable mutations.

3. Information on other populations, such as Asians, has not been included, so, the comprehensive picture of difference in gene alterations according to population is unclear; i.e., the authors simply showed a difference between African Americans and European Americans.

We thank the reviewer for this point and also think that assessing other populations for these mutations would be interesting. It was not possible for us to include Asian populations in our study as the populations recruited into the NCI-MD case control study are primarily of European and African ancestry and we did not have access to tissues from Asian American patients. In TGCA LUAD, there are 8 patients classified as Asian. Of these, one had a mutation in PTPRT (13%) and none had a mutation in JAK2, suggesting, albeit with low power, that the frequency of PTPRT mutations among Asian American populations is more consistent with that of European Americans. As Asian Americans are one of the fastest growing racial groups in the US, we agree that assessing the somatic mutation profile in this population is also very important. It was, however, outside the scope of our study. We added a discussion of this point on page 11. We also analyzed non-USA data from ICGC and there, in the Korean population, found a mutation rate of 5% among LUSC patients. There were no LUAD data from non-US populations in this database.

Page 10/11:

Whether or not population differences in PTPRT/JAK2 mutations extend to populations of Asian descent remains to be determined. TCGA has eight LUAD patients classified as Asian, one (12.5%) of which carries a PTPRT mutation, suggesting that the frequency is more closely aligned with European Americans.

4. The reviewer does not fully agree that African Americans are called as a minority.

Presently, US Census Bureau statistics estimate that 13.4% of the U.S. population is Black or African American. The US government, Census Bureau and National Cancer Institute consider African Americans to be a minority and under-represented population, which is why we have classified African Americans as such in our manuscript. Further, our use of the term “underrepresented” in this manuscript refers to the fact that African Americans are not proportionally or sufficiently (from a power perspective) represented in genomics research, an important research gap that we have tried to address.

Reviewer #2 (Remarks to the Author):

This study reports some distinct alterations in PTPRT/JAK2 mutation in lung adenocarcinoma among African Americans. The strengths are a) study focused on a population often overlooked in genomic studies b) confirmation from two different cohorts including one from TCGA. I have a few comments

1. The abstract should include more details on the initial cohort and validation cohort

We thank the reviewer for this comment and are happy to address it. Please see below the additional details we added to the abstract on page 2.

Page 2

Despite persistent disparities in lung cancer incidence and survival, our current understanding of lung cancer biology is derived primarily from populations of European descent. We initially used a targeted sequencing panel and patient samples from the NCI-MD case control study where we identified significantly higher prevalence of PTPRT and JAK2 mutations in lung adenocarcinomas among African Americans compared with European Americans. This increase in mutation frequency was validated with independent WES data from the NCI-MD study and TCGA. We found that patients carrying these mutations had a concomitant increase in IL-6/STAT3 signaling and miR-21 expression. The identification of these potentially actionable mutations could have clinical significance for targeted therapy and the enrollment of minority populations in clinical trials.

2. The text should highlight whether mismatch repair genes were altered in the hypermutated samples

We agree with the reviewer that this point should have been mentioned. *MSH2, MSH6, MLH1* and *PMS2* were included on the targeted panel. One of the tumors from a hypermutated sample had *MSH2* mutation. We have now added these details to page 4.

Page 5

Mutations in mismatch repair genes have been associated with a hypermutated phenotype. MSH2, MSH6, MLH1, and PMS2 were included in the gene panel and only one of the NCI-MD patients had a mutation, which was a missense R638S mutation in MSH2.

3. It is important to be careful in discussing inter tumor heterogeneity when multiple samples from a given lesion have not been studied and the panel of genes tested was limited at least in the initial discovery cohort (page 4)

We appreciate the reviewer's comment here and they are correct. Our goal was to highlight heterogeneity that exists among samples from different individuals as compared with samples from the same individual. We have therefore modified the text on page 4 to make this point clearer, which we hope is now the case.

Page 5

*Consistent with published studies, we observed a complex mutational landscape of lung cancer in AAs with considerable **heterogeneity in the somatic landscape between individuals** and little evidence for dominant driver genes (Supplementary Fig. 1C).*

4. I would delete the part discussing CSMD3, RYR2 and SYNE1- they are not considered relevant in this context as the authors state

We agree with this comment and are happy to delete it.

5. In the absence of functional studies, it is a leap of faith to say that "our findings suggest that patients carrying these mutations may be more likely to respond to such drugs"- This sentence should be modified

We agree with the reviewer that we should modify the sentence to more accurately reflect the *hypothesis* that individuals carrying these mutations could be more likely to respond to drugs as opposed to saying that the results suggest that patients *would* more likely respond and have made these changes on page 10. We have also conducted an analysis using cell line drug sensitivity data to support the hypothesis that these mutations could be actionable.

Page 9/10

*These data suggest that an increased frequency of loss of function *PTPRT* and *JAK2* mutations may drive *STAT3* activity in subsets of NSCLC that are enriched among AAs. Interestingly, our recent work demonstrated that while *IL-6* is associated with lung cancer diagnosis in both EAs⁵ and AAs⁶, the effect size was considerably larger among AAs, which is further evidence that this *IL-6/JAK2/STAT3* pathway is important among AAs. We hypothesize that patients with *PTPRT* and *JAK2* mutations could be candidates for targeted therapy and as such, our findings have implications for the recruitment of patients into clinical trials. **For example, the initial***

conception to use JAKs as therapeutic targets was based on the identification of an activating mutation in JAK2 linked to myeloproliferative neoplasms². The rationale for their use in these disorders has also been linked with perturbed JAK/STAT signaling, either due to somatic mutations or transcriptomic changes⁷. Current JAK inhibitors are not always selective and most do not target specific mutations, though newer generations of JAK inhibitors demonstrate selective inhibition. JAK2 inhibitors might not work in PTPRT mutant tumors because other JAKs can, in theory, activate STAT3. As such, STAT3 inhibitors are good candidates for the tumors we describe in our study. Interestingly, we conducted an agnostic analysis of differential drug sensitivity among cell lines mutant for JAK2 or PTPRT using the depmap database⁴ (<https://depmap.org>) and identified a STAT inhibitor (Supplementary Table 11). Our findings therefore raise the hypothesis that patients carrying these mutations may be more likely to respond to drugs that target this pathway than patients without these mutations. However, detailed mechanistic experiments will be needed to determine whether these are indeed actionable mutations.

6. It is not correct to state this is the first study to examine genomic alterations in AA

We completely agree with the reviewer and we frequently referenced these previous papers in the manuscript. What we had written is that our study is the first analysis of matched tumor and normal frozen tissues from African Americans, which, to our knowledge, is correct. There was a study of matched tumor and normal samples earlier this year, but the study used DNA from FFPE samples⁸. We highlighted the inclusion and use of fresh frozen tissues as this tissue type can give higher quality calls and results for next gen sequencing. That aside, we accept the reviewer's point that this may not be immediately clear, and as such, we have modified the statement on page 10.

Page 10

Our study has several strengths. It uses fresh frozen tissues and matched tumor and non-involved adjacent tissues.

7. There is not much discussion on the alterations (if any) that are unique to African ancestry not captured in the current reference genome

The reviewer makes a valid and important point here. The work that the reviewer refers to is recent data showing that the African pan-cancer genome contains ~10% more DNA than is captured in the current reference genome⁹ (which was primarily derived from an individual of European descent). The paper and similar work has led to a discussion of whether or not the field needs to move from using one reference genome to population-specific reference genomes. It is not clear exactly which coding regions that the novel 10% maps to (it is a fraction), but the point is still a very valid one as both normal and tumor samples are still mapped to the reference genome and as such, any "novel" DNA sequences would have been thrown out. This is an important discussion point and we have now expanded on it on page 10.

Page 10

This study design gives us the ability to call true somatic mutations and is especially important in light of recent findings suggesting that up to 10% of the genome in individuals of African

ancestry are not captured using the current reference genome⁹. *Most of these differences map to intergenic and non-coding regions, as such, their impact on a targeted exome sequencing panel would be expected to be limited in nature. However, future work should address these novel genomic sequences and assess them for potential health-associated variants.*

References

1. Raphael BJ. Chapter 6: Structural variation and medical genomics. *PLoS Comput Biol* **8**, e1002821 (2012).
2. Baxter EJ, *et al.* Acquired mutation of the tyrosine kinase JAK2 in human myeloproliferative disorders. *Lancet* **365**, 1054-1061 (2005).
3. Schust J, Sperl B, Hollis A, Mayer TU, Berg T. Stattic: a small-molecule inhibitor of STAT3 activation and dimerization. *Chem Biol* **13**, 1235-1242 (2006).
4. Corsello SM, *et al.* Non-oncology drugs are a source of previously unappreciated anti-cancer activity. *bioRxiv*, 730119 (2019).
5. Brown D, *et al.* Relationship between circulating inflammation proteins and lung cancer diagnosis in the National Lung Screening Trial. *Cancer Epidemiol Biomarkers Prev*, (2018).
6. Meaney CL, *et al.* Circulating Inflammation Proteins Associated With Lung Cancer in African Americans. *J Thorac Oncol*, (2019).
7. Garbers C, Heink S, Korn T, Rose-John S. Interleukin-6: designing specific therapeutics for a complex cytokine. *Nat Rev Drug Discov* **17**, 395-412 (2018).
8. Lusk CM, *et al.* Profiling the Mutational Landscape in Known Driver Genes and Novel Genes in African American Non-Small Cell Lung Cancer Patients. *Clin Cancer Res*, (2019).
9. Sherman RM, *et al.* Assembly of a pan-genome from deep sequencing of 910 humans of African descent. *Nature genetics*, (2018).

REVIEWERS' COMMENTS:

Reviewer #1 (Remarks to the Author):

The authors have not examined oncogenic fusions, so, it still remains unclear whether PTPRK/JAK2 mutations are driver alterations or not. However, the authors carefully revised the manuscript, so, the readers' over-interpretation will be avoided. The reviewer hopes that this publication will proceed to target therapies for AA patients.

Reviewer #2 (Remarks to the Author):

The authors have addressed all of my concerns very appropriately. The revised manuscript reads very well and will be an important paper addressing an often overlooked population with regard to genomic alterations in cancer.